# Treatment outcomes among young persons living with HIV who transitioned to adult care in southern Nigeria: A retrospective cohort study

Esther Nwanja[1]ᴥ*, Uduak Akpan[1]ᴥ, Otoyo Toyo[1]ᴥ, Ogheneuzuazo Onwah[1]ᴥ, Chinazom Ekwueme[2], Ofonime Dixon-Umo[3], Bala Gana[1], Godson Ndubueze[4], Ijeoma Uchenna Itanyi[2], Echezona Ezeanolue[2], Augustine Idemudia[5], Onyeka Igboelina[5], Dolapo Ogundehin[5], Ezekiel James[5], Chika Obiora-Okafo[5], Olugbenga Asaolu[5], Bayo Onimode[5], Moses Katbi[5], Jemeh Pius[5], Omosalewa Oyelaran[5], Okezie Onyedinachi[1], Adeoye Adegboye[1], Andy Eyo[1]

**1** Excellence Community Education Welfare Scheme, Uyo, Nigeria, **2** Center for Translation and Implementation Research, College of Medicine, University of Nigeria, Enugu, Nigeria, **3** University of Uyo Teaching Hospital, Uyo, Nigeria, **4** Johns Hopkins Centre for Public Health and Human Rights, Maryland, United States of America, **5** Office of HIV and TB, United States of America Agency for International Development (USAID), Abuja, Nigeria

ᴥ These authors contributed equally to this work
* airstarnwanja@gmail.com/enwanja@ecews.org

## Abstract

### Background

In October 2019, a peer-based transition preparedness model was introduced as part of peer club activities to prepare young persons living with HIV (YLHIV) for adult care. This study compared the 12 and 24 months treatment outcomes of YLHIV who transitioned to adult care in primary, secondary and tertiary health facilities in Southern Nigeria, following the introduction of this model.

### Materials and methods

This was a retrospective cohort study using data extracted from the medical records of YLHIV who transitioned to adult care at 25 years in 2018 and in 2021 across 155 healthcare facilities in southern Nigeria. Baseline data at transition, as well as 12 and 24 months post-transition data were extracted for comparison between those who were transitioned before (2018 cohort) and after (2021 cohort) the transition preparedness model was introduced. Logistics regression analysis was used to compare client continuity on treatment and undetectable viral load between the two groups at 12 and 24 months after transitioning to adult care.

### Results

Most of the 1,555 YLHIV who transitioned to adult care in 2018 (n=343, 22.1%) and 2021 (n=1,212, 77.9%) were females (91.0% in 2018 v.82.6% in 2021) and initiated

**Data availability statement:** All relevant data are within the manuscript and its Supporting Information files.

**Funding:** This study was supported by the United States Agency for International Development (grant number 72062022CA00007 to ECEWS).

**Competing interests:** The authors have declared that no competing interests exist.

ART at 20 years or older (92.7% v. 95.7%). A higher proportion of those in the 2021 cohort were continuously retained both at 12 months and 24 months post-transitioning compared to those in the 2018 cohort (12 months: 96.7% vs 80.2%, p<0.001; 24 months: 92.7% vs 77.6% p<0.001). Similarly, the proportion of YLHIV with undetectable viral load in the 2021 cohort was significantly higher than those in the 2018 cohort at both 12 months (96.1% vs 60.1%, p<0.001) and 24 months (93.3% vs 80.6%, p<0.001), respectively.

## Conclusion

Peer-based transition preparedness model improved treatment outcomes of YLHIV who transition to adult care. Programs should implement tailored, peer-based interventions to address gaps in service delivery.

## Introduction

Life expectancy among people living with HIV has improved since the introduction of antiretroviral therapy (ART) [1,2]. AIDS-related deaths have generally halved since 2010 [3], and perinatally infected children are now growing from adolescence to adulthood [4–6]. However, AIDS-related deaths among young persons living with HIV (YLHIV) have only marginally reduced over the last decade [7]. Sub-Saharan Africa bears the brunt of the HIV epidemic, accounting for 89% of all adolescents living with HIV worldwide [8]. Nigeria has the largest population of adolescents living with HIV in West and Central Africa [8] with less than 90% currently receiving treatment [9].

YLHIV (adolescents aged 10–19 and youth aged 15–24) face a critical period when transitioning from adolescent to adult HIV care [10]. This transition often coincides with significant physiological, psychological, and social changes [10]. Additionally, this transition may involve changes in clinic settings, models of care, and healthcare providers. These factors can increase the risk of negative treatment outcomes, such as reduced adherence and treatment interruptions [11,12]. Davies et al who tracked the outcomes of four cohorts of adolescents who transitioned their care to another facility in the Western Cape Province of South Africa noticed a decline in the post-transition viral suppression rate [13]. Meloni et al also observed high post-transition treatment attrition among adolescents who transitioned from paediatric HIV clinics to adult clinics in Nigeria [14].

Several models of care have been developed to support YLHIV during this transition [15–18]. These include integrated family-centred approaches [15], teen clubs [16], transition camps [17], and youth-friendly transition clinics [18]. Some countries, including Nigeria, have introduced guidelines on interventions to support YLHIV during this transition phase [15,19]. In October 2019, a peer-based transition preparedness model was introduced as part of peer club activities in southern Nigeria. This model, funded by the United States President's Emergency Plan for AIDS Relief (PEPFAR) through the United States Agency for International Development (USAID), differed from other adolescent-focused programs. While existing programs primarily

addressed treatment adherence and psychosocial support during adolescence, this model specifically prepared YLHIV for long-term adult care. It delivered peer-based services tailored to the age group, age-appropriate transition information, psychosocial support, and adherence support.

Evaluations of these interventions have yielded mixed results. Hansudewechakul et al. reported improved viral suppression among transitioning adolescents in Thailand using a group transition system [17]. Lolekha et al. found improvements in HIV and health-related knowledge among perinatally infected youths (aged 14–22) using an 18-month "Happy Teen Programme" in Thailand [16]. Foster and colleagues utilised a multidisciplinary youth-friendly transition intervention in a tertiary setting in London [20]. However, Brain et al. found lower retention among adolescents who transitioned to the adult clinic compared to those who remained in the paediatric clinic in South Africa [21]. In Thailand, a 12- and 24-month evaluation of YLHIV who transitioned to adult care found suboptimal and comparable treatment outcomes among participants in a teen program and those who did not participate [22].

While existing studies offer valuable insights into treatment outcomes for YLHIV transitioning to adult care, they have focused mainly on perinatally infected adolescents and often involve specialised or tertiary healthcare facilities. Consequently, there is a paucity of evidence on the post-transition outcomes for YLHIV in other facility types, such as primary and secondary care facilities, particularly in resource-limited settings like Nigeria. This study addresses this gap by examining the treatment outcomes among YLHIV who transitioned to adult care using a peer-based transition preparedness model at different facility types in Nigeria.

The objective of this study was to assess the 12 and 24-month post-transition continuity on treatment and viral suppression rates among YLHIV who transitioned to adult HIV care at primary, secondary and tertiary facilities in southern Nigeria. The study considered the outcomes before and after implementing the peer-based transition preparedness model, including client demography and facility types.

## Materials and methods

### Study design and population

This was a retrospective before-and-after cohort study of YLHIV who transitioned to adult care. The study included two cohorts: a 2018 cohort (pre-intervention) who transitioned before the introduction of a transition preparedness model and a 2021 cohort (post-intervention) who transitioned after the model's implementation. Both cohorts transitioned at 25 years of age at health facilities in Akwa Ibom and Cross River States, Nigeria. The study assessed the model's effectiveness by comparing post-transition treatment outcomes between the two cohorts. The retrospective before-and-after cohort design was chosen because it allows for a direct comparison of outcomes to determine the effect of the intervention since this was part of program implementation. All PEPFAR/USAID-supported facilities in these two states were included, and all YLHIV within these facilities who met the pre-specified inclusion criteria were included.

### Study setting

The study was conducted in Akwa Ibom and Cross River States. These states were selected for this study because they collectively account for 12% of Nigeria's national HIV burden [23] and received funding from the PEPFAR through USAID to implement the intervention.

The study was conducted in all 155 PEPFAR-supported health facilities which include 104 primary, 49 secondary and 2 tertiary health facilities. The primary health facilities are widely distributed, located closer to communities, and are often the first point of contact with healthcare for most people [24] They are often staffed by nurses and community health workers who provide general care [25]. Secondary health facilities serve as referral centres for primary health facilities and offer limited specialised services provided by doctors and nurses. Tertiary health facilities, which serve as the highest level of referrals, have highly experienced experts offering specialised care for disease conditions [25,26].

In PEPFAR-supported primary and secondary health facilities in Akwa Ibom and Cross River states, clinicians employed through USAID funding provide technical assistance to service providers on HIV management.

## Pre-intervention Period

Before October 2019, YLHIV did not receive any specific intervention to prepare them for the transition to adult care. In primary and secondary healthcare facilities, YLHIV received care in the same clinics and from the same healthcare providers as adults. However, in tertiary health facilities, YLHIV received ART services in paediatric clinics until they transferred to adult clinics at 18 years. Case managers provided treatment support to all clients, including YLHIV, however, this was not tailored to any subpopulation.

## Description of the Intervention

Beginning in October 2019, a peer-based transition preparedness model was implemented as part of newly introduced facility-based adolescent peer club activities. Table 1 summarises the essential features of this model. The model aimed to prepare YLHIV for adult care. All YLHIV in care whose HIV status had been fully disclosed to them were offered membership in the peer-based clubs, which were organised by age bands, 10–14 years, 15–19 years, and 20–24 years. YLHIV who joined the clubs graduated from one age band to the next as they grew older and exited the program at 25 years of age. The clubs met monthly, moderated by trained peer facilitators with support from the health facility's clinical team. These meetings also served as service delivery points for medication pickups, psychosocial support, viral load [VL] sample collection, and other services.

The peer facilitators, who moderated the group meetings, were purposively selected from the treatment-experienced YLHIV at each facility. These were individuals who showed good self-care and were willing to disclose their HIV status to their peers and support them on their ART journey. They were then trained to provide adolescent-friendly services. During club meetings, they provided age-appropriate literacy materials and orientation on self-management; shared experience on ART, one-on-one support sessions to YLHIV (even outside peer club meetings), and psychosocial support. During the COVID-19 lockdown, virtual meetings replaced in-person club meetings. In addition to the clubs, all YLHIV were assigned to adolescent-focused case managers trained in adolescent-friendly care who provided continuous treatment support to YLHIV.

**Table 1. Outline of the peer-based transition preparedness model.**

| Who | • **Trained peer facilitators, Health facility clinical team** |
|---|---|
| Where | • Health facility |
| When | • Monthly (although individuals attended quarterly) |
| What | • Identify subpopulation for transition (at 15–24 years)<br> ◦ Notification of the timeline for transitioning<br> ◦ Transition readiness assessment<br>• Transition Preparation<br> ◦ Education<br> • Treatment Literacy: HIV, ART, Adherence and importance, ART adherence, Viral load monitoring.<br> • Sexual and reproductive health<br> • Disclosure to sexual partners and treatment supporters<br> • Peer-to-peer support<br> • Exiting adolescent-based care<br> ◦ Establishment of a one-on-one relationship for adherence and psychosocial support between YLHIV and peer facilitators.<br>• Transfer to adult care/service provider (at 25 years)<br> ◦ Discontinue facility-based peer club activities<br> ◦ Reassignment to an adult case manager and introduction of YLHIV to the case managers in the adult clinic |

The peer-based transition preparedness model had four essential components. First was orienting YLHIV on the age of transition. This was done from the age of 15 years. YLHIV received orientation at club meetings about the transition to adult care at 25 years of age. The second component was a transition readiness assessment where healthcare workers routinely assessed YLHIV's knowledge needs through informal discussions during clinic visits and club meetings and used this information to tailor support services. The third component was transition preparedness using a six-month transition curriculum. This curriculum covered topics on self-care, roles, responsibilities, expectations during and after the transition, treatment literacy, sexual and reproductive health, disclosure to partners, and peer support. Routine refreshers were provided at age-band groupings. The fourth was the actual transfer to adult care, where YLHIV were disengaged from club activities and were assigned/physically introduced to adult case managers when they turned 25 years old. They were also linked to Differentiated service delivery (DSD) for adults [27]

## Data collection and variable definitions

Data were extracted from the electronic medical records of YLHIV who transitioned to adult care in 2018 and 2021. YLHIV were considered to have transitioned upon reaching 25 years of age.

**Inclusion criteria.**

• YLHIV who turned 25 in 2018 and 2021.

• YLHIV enrolled in peer-based clubs (2021 cohort only)

**Exclusion criteria.**

• YLHIV who transferred out of the facility after transition were excluded because their treatment outcomes could not be established during the follow-up period.

• YLHIV with less than one year of HIV treatment before the transition were excluded from the 2021 cohort due to their limited exposure to the intervention.

• YLHIV not enrolled in the peer-based clubs (2021 cohort only).

**Variables.** Data were collected at baseline (25 years) and after 12 and 24 months post-transition. This included demographic data (age and sex), healthcare facility type (primary, secondary, and tertiary), and clinical data (year of ART initiation, antiretroviral [ARV] regimen at ART initiation and transition [categorised into Dolutegravir-based and non-Dolutegravir-based ART], viral load results with dates and ART treatment status).

The outcomes compared between the 2018 and 2021 cohorts include continuity on treatment (the proportion of YLHIV who were alive and receiving care), and undetectable viral load (the proportion of YLHIV with viral load <50 copies/ml) at 12 and 24 months post-transition. Continuity on treatment was disaggregated based on the frequency and duration of treatment interruption into "continuously retained" (no treatment interruptions throughout the period); "retained with treatment interruptions" (≥1 treatment interruption but active at the end of the period); and "not retained" (inactive at the end of the period). Viral load was categorised as sustained undetectable viral load (undetectable viral load [<50 copies/ml] at both 12 and 24 months), low-level viremia (≥1 viral load between 50 and 999 copies/ml during the period), and high viral load (any viral load ≥1000 copies/ml during the period).

## Data analysis

Descriptive statistics (frequencies and percentages) were used to summarise the client characteristics, disaggregated by transition cohort. Client continuity on treatment and undetectable viral load were determined at 12 months and 24 months post-transitioning and compared by transition cohorts using chi-square statistics. The various disaggregates for continuity on treatment and viral load at 12 and 24 months were also compared across transition periods using chi-square statistics.

Binary logistics regression analysis was performed to estimate the association between the treatment outcomes and the transition cohorts at 12 and 24 months. The logistic regression model was adjusted for age at transition, sex, and health facility type. These potential confounders were identified based on prior literature and clinical relevance. For the regression model, the 2018 cohort served as the reference category, and models were not analyzed if a variable had only one outcome category. All variables included in the logistic regression model had complete data, and no imputation methods were required. All analyses were performed using SPSS version 26, with the p-value set at 0.05. The reporting of this study conforms to the Strengthening the Reporting of Observational Studies in Epidemiology (STROBE) statement [28].

## Ethical approval

Permission to analyse the secondary program data was obtained from the Health Research Ethics Committee (HREC) in Akwa Ibom State on December 22, 2022 (HREC No. AKHREC/25/11/22/120), with an extension granted on January 9, 2024 (HREC No. AKHREC/5/5/23/149). Additional approval was obtained from the Office of International Research Ethics (OIRE) on February 24, 2023 (Project #: 2026278–1). Data was accessed from the electronic medical records on June 25, 2024. Informed consent was waived because the study data was collected retrospectively, and only de-identified data were used for the analysis.

## Results

The records of 1,555 YLHIV receiving first-line ART who transitioned to adult care in the two periods were reviewed (343 in the 2018 transition cohort and 1,212 in the 2021 transition cohort). The majority of the YLHIV in these transition cohorts were female (91.0% in the 2018 cohort, 82.6% in the 2021 cohort). Most persons initiated ART between ages 20 and 24 (92.7% in the 2018 cohort, 95.7% in the 2021 cohort). Regarding health facility type, 43.4% received services from primary healthcare facilities, 51.3% from secondary healthcare facilities, and 5.3% from tertiary healthcare facilities in the 2018 cohort, compared to 66.7%, 28.6%, and 4.7%, respectively, in the 2021 cohort. Table 2 below summarises the characteristics of the two transition cohorts.

**Table 2. Characteristics of YLHIV transitioned to adult care in 2018 and 2021 in Akwa Ibom and Cross River States, Nigeria (n = 1555).**

| | | 2018 | | 2021 | |
|---|---|---|---|---|---|
| | | (n = 343) | (%) | (n = 1212) | (%) |
| Sex | | | | | |
| | Male | 31 | 9.0% | 211 | 17.4% |
| | Female | 312 | 91.0% | 1001 | 82.6% |
| Age at ART initiation | | | | | |
| | 0 -14 years | 2 | 0.6% | 8 | 0.7% |
| | 15 - 19 years | 23 | 6.7% | 44 | 3.6% |
| | 20 -24 years | 318 | 92.7% | 1160 | 95.7% |
| Health facility type | | | | | |
| | Primary | 149 | 43.4% | 808 | 66.7% |
| | Secondary | 176 | 51.3% | 347 | 28.6% |
| | Tertiary | 18 | 5.3% | 57 | 4.7% |
| Regimen at ART Initiation | | | | | |
| | Dolutegravir based | 0 | 0% | 1,002 | 82.7% |
| | Non-Dolutegravir based | 343 | 100% | 210 | 17.3% |
| Regimen at Transition | | | | | |
| | Dolutegravir based | 2 | 0.6% | 1,209 | 99.8% |
| | Non-Dolutegravir based | 341 | 99.4% | 3 | 0.2% |

## ART regimen

All YLHIV who transitioned to adult care in 2018 initiated ART with non-Dolutegravir-based ART, with only 0.6% (n = 2) using Dolutegravir-based ART at the transition to adult care. In contrast, among the 2021 transition cohort, 82.7% (1,002) initiated ART on a Dolutegravir-based regimen, increasing to 99.8% (1,209) at the transition to adult care "Table 2".

## Continuity on treatment

As shown in Table 3, post-transition continuity on treatment at 12 months was higher for those in the 2021 transition cohort compared to those in the 2018 cohort (96.8% vs 93.6%, p = 0.006). However, post-transition continuity on treatment at 24 months was lower for those in the 2021 transition cohort compared to those in the 2018 transition cohort (95.0% vs 99.7%, p = 0.021).

Also, a higher proportion of those in the 2021 transition cohort were continuously retained both at 12 months and 24 months post-transitioning compared to those in the 2018 transition cohort (12 months: 96.7% vs 80.2%, p < 0.001; 24 months: 92.7% vs 77.6% p < 0.001).

As seen in Table 4, post-transition continuity on treatment at 12 months was higher among female YLHIV in the 2021 transition cohort compared to those in the 2018 transition cohort (97.4% vs 93.3%, OR: 2.71 95%CI [1.50–4.88]). Continuity on treatment at 12 months was also higher among those aged 20–24 years in the 2021 transition cohort compared to those in the 2018 transition cohort (97.0% vs 93.1%, OR: 2.39, 95%CI [1.381–4.134]). With regards to facility type, continuity on treatment at 12 months was higher among YLHIV receiving care in primary health facilities in the 2021 transition cohort compared to those in the 2018 cohort (98.8% vs 92.6%, OR: 6.36, 95% CI [2.651–15.262]).

At 24 months, post-transition continuity on treatment was lower for YLHIV aged 20–24 years in the 2021 transition cohort compared to those in the 2018 transition cohort (95.2% vs 99.7%, OR: 0.44, 95%CI [0.200–0.983]). By facility type,

**Table 3. Post-transition continuity on treatment at 12 and 24 months for YLHIV who transitioned to adult care in 2018 and 2021 in Akwa Ibom and Cross River States, Nigeria.**

| Characteristics | | 2018 transition cohort | 2021 transition cohort | $\chi^2$ | p-value |
|---|---|---|---|---|---|
| | | n = 343 | n = 1212 | | |
| By Period Category | | | | | |
| 12 months | | | | | |
| | Retained | 321 (93.6%) | 1174 (96.8%) | 7.74 | 0.006 |
| | Not retained | 22 (6.4%) | 38 (3.2%) | | |
| 24 months | | | | | |
| | Retained | 336 (99.7%) | 1151 (95.0%) | 5.72 | 0.021 |
| | Not retained | 7 (0.6%) | 61 (5.0%) | | |
| By continuity on treatment Category | | | | | |
| 12 months | | | | | |
| | Continuously retained | 275 (80.2%) | 1172 (96.7%) | 167.25 | <0.001 |
| | Retained with treatment interruption | 46 (13.4%) | 2 (0.2%) | | |
| | Not retained | 22 (6.4%) | 38 (3.1%) | | |
| 24 months | | | | | |
| | Continuously retained | 266 (77.6%) | 1124 (92.7%) | 172.13 | <0.001 |
| | Retained with treatment interruption | 70 (20.4%) | 27 (2.3%) | | |
| | Not retained | 7 (2.0%) | 61 (5.0%) | | |

(chi-square statistics)

**Table 4. Post-transition continuity on treatment at 12 and 24 months for YLHIV who transitioned to adult care in 2018 and 2021, in Akwa Ibom and Cross River States, Nigeria, disaggregated by demographic characteristics.**

| Characteristics | 12 months post-transition | | | | | | | 24 months post-transition | | | | | | |
|---|---|---|---|---|---|---|---|---|---|---|---|---|---|---|
| | 2018 Transition cohort | | | 2021 Transition cohort | | | Adjusted Odds Ratio [95% CI] | 2018 Transition cohort | | | 2021 Transition cohort | | | Adjusted Odds Ratio [95% CI] |
| | Total YLHIV | YLHIV active on treatment | Continued treatment (%) | Total YLHIV | YLHIV active on treatment | Continued treatment (%) | | Total YLHIV | YLHIV active on treatment | Continued treatment (%) | Total YLHIV | YLHIV active on treatment | Continued treatment (%) | |
| Total | 343 | 321 | 93.6% | 1212 | 1174 | 96.9% | 2.12 [1.23–3.63]* | 343 | 336 | 98.0% | 1212 | 1151 | 95.0% | 0.39 [0.178–0.867]* |
| Sex | | | | | | | | | | | | | | |
| Male | 31 | 30 | 96.8% | 211 | 199 | 94.3% | 0.55 [0.069–4.41] | 31 | 31 | 100.0% | 211 | 197 | 93.4% | NA |
| Female | 312 | 291 | 93.3% | 1001 | 975 | 97.4% | 2.71 [1.50–4.88]* | 312 | 305 | 97.8% | 1001 | 954 | 95.3% | 0.47 [0.208 - 1.041] |
| Age at ART initiation | | | | | | | | | | | | | | |
| 0 -14 years | 2 | 2 | 100.0% | 8 | 8 | 100.0% | NA | 2 | 2 | 100.0% | 6 | 6 | 100.0% | NA |
| 15 - 19 years | 23 | 23 | 100.0% | 44 | 41 | 93.2% | NA | 23 | 23 | 100.0% | 44 | 39 | 98.0% | NA |
| 20 - 24 years | 318 | 296 | 93.1% | 1160 | 1125 | 97.0% | 2.39 [1.381–4.134] * | 318 | 311 | 99.7% | 1160 | 1104 | 95.2% | 0.44 [0.200–0.983]* |
| Health facility type | | | | | | | | | | | | | | |
| Primary | 149 | 138 | 92.6% | 808 | 798 | 98.8% | 6.36 [2.651 - 15.262]* | 149 | 146 | 98.0% | 808 | 783 | 96.9% | 0.64 [0.192–2.159] |
| Secondary | 176 | 165 | 93.7% | 347 | 328 | 94.5% | 1.15 [0.535–2.475] | 176 | 172 | 97.7% | 347 | 320 | 92.2% | 0.276 [0.095–0.801]* |
| Tertiary | 18 | 18 | 100.0% | 57 | 48 | 84.2% | NA | 18 | 18 | 100.0% | 57 | 48 | 84.2% | NA |

\* statistics based on logistic regression analysis; p-value <0.05; reference category is 2018; NA means one predictor has a single outcome and thus is not analyzed in a model.

**Table 5. Proportion of YLHIV with undetectable viral load at 12 and 24 months after transitioning to adult care in 2018 and 2021, and viral load changes over the 24 months post transitioning, in Akwa Ibom and Cross River States, Nigeria.**

| Characteristics | Proportion of YLHIV with undetectable VL | | $\chi^2$ | p-value |
|---|---|---|---|---|
| | **2018** | **2021** | | |
| By year* | | | | |
| 12-month | 197 (60.1%) | 1122 (96.1%) | 84.34 | <0.001 |
| 24-month | 175 (80.6%) | 1055 (93.3%) | 14.77 | <0.001 |
| By VL category** | | | | |
| Sustained undetectable viral load | 97 (48.0%) | 1000 (90.1%) | 45.75 | <0.001 |
| Low-level viral load | 57 (28.2%) | 86 (7.7%) | 19.16 | <0.001 |
| High viral load | 48 (23.8%) | 24 (2.2%) | 46.14 | <0.001 |

*denominator includes only those with viral load results within the period;

**only for those with at least two viral load results post-transition over 24 months; 2018 = 202; 2021 = 1110. Sustained undetectable viral load means viral load ≤50 copies/ml, low-level viral load means ≥1 viral load between 51 and 999 copies/ml, and high viral load means any viral load ≥1000 copies/ml during the period.

continuity on treatment at 24 months was lower among YLHIV receiving care in secondary health facilities in the 2018 transition cohort compared to those in the 2021 cohort (92.2% vs 97.7%, OR: 0.276, 95%CI [0.095–0.801]).

## Viral load suppression

The proportion of YLHIV with undetectable viral load after transitioning to adult care was higher among those in the 2021 transition cohort compared to those in the 2018 cohort both at 12 months (96.1% vs 60.1%, (p < 0.001) and 24 months (93.3% vs 80.6%, p < 0.001) respectively. Similarly, 90.1% of YLHIV in the 2021 transition cohort had sustained undetectable viral load compared to 48.0% in the 2018 cohort (p < 0.001) "Table 5".

The proportion of YLHIV with undetectable viral load was higher at 12 months in the 2021 transition cohort in all demographic categories except females (96.3% vs 60.9%, OR: 2.52, 95% CI [2.207–2.886]), and also at 24 months in all demographic categories except YLHIV receiving care in tertiary health facilities (85.1% vs 87.5%, OR 0.816, 95%CI [0.151–4.403]), compared to the 2018 year "Table 6".

## Discussion

This study found that continuity on treatment after the introduction of a peer-based transition preparedness model (2021 transition cohort) was higher at 12 months for YLHIV compared to before (2018 transition cohort) but lower at 24 months for the 2021 cohort. The proportion of YLHIV with undetectable viral load at 12 and 24 months post-transitioning was also significantly higher after the introduction of the model.

Continuity on treatment in adulthood is a recognized global challenge for YLHIV. In Nigeria, YLHIV face unique challenges such as stigma, lack of psychosocial support (especially among older adolescents who may be in boarding schools or residing in universities), financial stress, and disruptions in care due to the transition from paediatric to adult services, which can impact treatment adherence and outcomes [11,12,14,29]. The introduction of a peer-based intervention aimed to mitigate these issues by providing a supportive community and structured preparation for transition [14].

YLHIV who passed through the peer-based transition model had retention rates that reached the UNAIDS 95% target for sustained ART [30,31] at 12 months (96.8%) and 24 months (95.0%) post-transition. Our findings were similar to those from a cluster randomised control trial among youth (15–24 years) with HIV in Kenya [15], where retention was 97% at 12 months post-transition. An adolescent transition package was used to support transition readiness just like in our setting, which could explain the similarity in retention outcomes. In contrast, a longitudinal study among adolescents aged 10–19

**Table 6. Proportion of YLHIV with undetectable viral load at 12 and 24 months after transitioning to adult care in 2018 and 2021 in Akwa Ibom and Cross River States, Nigeria, disaggregated by demographic characteristics.**

| Characteristics | 12 months post-transition | | | | | 24 months post-transition | | | | |
|---|---|---|---|---|---|---|---|---|---|---|
| | 2018 transition cohort | | 2021 transition cohort | | Odds Ratio (95% CI) | 2018 transition cohort | | 2021 transition cohort | | Odds Ratio (95% CI) |
| | Number with undetectable VL | Undetectable VL (%) | Number with undetectable VL | Undetectable (%) | | Undetectable VL (%) | Number with undetectable VL | Number with undetectable VL | Undetectable VL (%) | |
| Total | 197/328 | 60.1% | 1122/1168 | 96.1% | 16.22 (11.221–23.445)* | 175/217 | 80.7% | 1055/1131 | 93.3% | 3.33 (2.212–5.017)* |
| Sex | | | | | | | | | | |
| Male | 16/31 | 51.6% | 194/203 | 95.6% | 2.72 (1.971–3.765)* | 14/20 | 70.0% | 178/194 | 91.6% | 4.768 (1.612–14.104)* |
| Female | 181/297 | 60.9% | 928/965 | 96.2% | 2.52 (2.207–2.886) | 161/197 | 81.7% | 877/937 | 93.6% | 3.27 (2.092–5.106)* |
| Age at ART initiation | | | | | | | | | | |
| 0 -14 years | 1/2 | 50.% | 7/8 | 87.5% | – | 1/1 | 100.0% | 7/8 | 87.5% | NA |
| 15 - 19 years | 16/23 | 69.6% | 40/43 | 93.0% | 5.83 (1.339–25.409)* | 13/16 | 81.3% | 38/40 | 95.% | 4.38 (0.658–29.221) |
| 20-24 years | 180/303 | 59.4% | 1075/1117 | 96.2% | 17.49 (11.911–25.683)* | 161/200 | 80.5% | 1010/1083 | 93.3% | 3.351 (2.196–5.115) |
| Health facility type | | | | | | | | | | |
| Primary | 87/138 | 63.0% | 754/781 | 96.5% | 16.37 (9.766–27.441)* | 85/105 | 81.0% | 725/771 | 94.0% | 3.708 (2.095–6.564)* |
| Secondary | 98/172 | 57.0% | 319/335 | 95.2% | 15.05 (8.380–27.047)* | 76/96 | 79.2% | 290/313 | 92.7% | 3.318 (1.731–6.358)* |
| Tertiary | 12/18 | 66.7% | 49/52 | 94.2% | 8.17 (1.781–37.449)* | 14/16 | 87.5% | 40/47 | 85.1% | 0.816 (0.151–4.403) |

*statistics based on logistic regression analysis; p-value <0.005; reference category is 2018; NA means one predictor has a single outcome and thus is not analyzed.

years at the Jos University Teaching Hospital, Nigeria, found that 87.7% of adolescents who transitioned to adult care at 18 years were retained in care 12 months post-transitioning [14], while Davies and colleagues in South Africa reported post-transition retention of 90% among adolescents aged 10–14 years after one year [13]. Both studies only evaluated transition outcomes among YLHIV who transitioned at tertiary healthcare facilities and neither study reported an intervention to support the transition process, which may be the reason for the lower retention rates reported. Our study shares a retrospective cohort design with Davies et al. [13] and Meloni et al.[14] but uniquely includes transitions across primary, secondary, and tertiary facilities in Nigeria. In the South Africa study, the majority of the transitions occurred before 15 years of age and involved a transition between healthcare facilities, whereas in our study, the transitions to adult care were at 25 years of age, and the majority of YLHIV remained at the same healthcare facility.

The success of the intervention in our study could be attributed to the peer-led support, which enhanced psychosocial resilience and adherence education, which enabled a supportive environment for YLHIV during transition [32,33]. The structured peer clubs and case management may have provided the tailored support that is crucial for maintaining high treatment retention settings [32–34]. Peer facilitators provide relatable guidance and psychosocial support, which may have increased adherence to ART and clinic visits [34,35], as evidenced by the higher proportion of YLHIV continuously retained in the 2021 cohort. Additionally, the structured transition preparation, including readiness assessments and peer mentoring, helps equip YLHIV with the knowledge and confidence needed for a successful transition to adult care [36]. These findings suggest that implementing similar peer-based models in other resource-limited settings could improve transition outcomes by addressing the social, psychological, and clinical needs of YLHIV.

Although most interventions and studies on adolescent transition have focused on tertiary health facilities [14,37], YLHIV in primary and secondary health facilities also eventually transition to adult care and therefore require support through their transition process [32]. The transitions at these sites may or may not involve a change in clinic or healthcare provider, but they often involve a change in the method of service delivery. This is a shift towards more direct, less mediated service delivery models where YLHIV take on greater responsibility for their care [11,15,33]. YLHIV at primary healthcare facilities in our setting had better retention at 12 months after the intervention compared to before, although the retention at 24 months was comparable before and after the intervention in this health facility type. The improved treatment continuity rate in primary health facilities suggests that the addition of the transition preparedness model strengthened the continuity of care for YLHIV. Participation in peer-led interventions that include peer interactive meetings, one-on-one support and adolescent-tailored service delivery has also been associated with improved adherence and continuity on treatment in other settings [34,35,38–40].

YLHIV who initiated ART at 20 years and older had a lower retention rate in the pre-intervention (2018 transition) cohort compared to those in the 2021 transition cohort. This is consistent with findings from Niwa et al., who reported a low retention rate for YLHIV who initiated ART in adolescence [39]. Also, Meloni et al. reported that the likelihood of treatment interruption was lower among those with longer ART duration prior to the transition [14]. The lower treatment continuity rate among young persons with shorter ART experience prior to transition has been attributed to several factors including the higher risk of sexual and gender-based stigma among those diagnosed with HIV in adolescence [39].

Our study found a higher proportion of YLHIV retained on treatment at 24 months in the 2018 transition cohort compared to those in the 2021 cohort. This might be due to the impact of COVID-19 in 2020, which coincided with the second-year post-transition for the 2018 cohort. COVID-19 restrictions led to the scale-up of innovations like differentiated service delivery of ART, including home refills, to sustain access to ART [27,41]. Semo et al. demonstrated that DSD improves client treatment retention, especially among adults, and the choice of a DSD model was influenced by clients' experience, cost, and convenience [42]. This effect of DSD may explain the higher treatment retention at 24 months among YLHIV in the 2018 cohort. In contrast, the proportion of YLHIV who were continuously retained was higher among those in the 2021 transition cohort compared to the 2018 cohort. The peer-based club into which the transition model was integrated emphasised adherence to clinic appointments and medications [36] and the higher tolerability of the Dolutegravir-based

regimen compared to other regimens [43] may account for the higher continuous retention among YLHIV in the 2021 transition cohort.

A higher proportion of YLHIV in the 2021 transition cohort had undetectable viral load at 12 and 24 months post-transition than in other studies [40,44]. For example, Meloni et al. [14], Griffith et al. [45] and Rungmaitree et al. [22] reported 34.6%, 67% and 73.3% viral suppression at 12 months respectively [14,22,45] and Rungmaitree et al. reported 76.7% viral suppression at 24 months post transitioning which are all below the 96.1% and 93.3% undetectable viral load which we reported at 12 months and 24 months post transitioning, respectively. These studies [14,22,45] attributed poor medication adherence among transitioned YLHIV to the poor suppression rate and where the ARV regimen was reported, YLHIV were predominantly receiving non-Dolutegravir based (NNRTI- or NRTI) regimen pre- or/and post-transition compared to our setting where YLHIV in the 2021 transition cohort predominantly received Dolutegravir-based regimen post-transitioning. Nigeria implemented changes in ART regimen for adults and children in 2018 with a full scale-up in 2019, recommending Dolutegravir-based medications as the preferred first-line regimen to replace the previously recommended Efavirenz-based regimen in adults and older children [46]. This is a likely contributor to the higher undetectable viral load in the 2021 transition cohort at 12 and 24 months, as Dolutegravir has been linked with similar or better tolerability, viral suppression, and undetectable viral load than non-Dolutegravir containing ART [43,47–51].

The higher sustained undetectable viral load among YLHIV who transitioned in the 2021 cohort could be attributed to sustained retention in care, which plays a key role in viral suppression [52]. Also, undetectable viral load was emphasised as a goal of the peer-based club, "zero viral load" [36]. The intervention reinforced this messaging and may have also contributed to the significantly higher undetectable viral load among YLHIV transitioned after the intervention.

### Limitations and strengths

Our study had some limitations. The use of secondary data limited our ability to consider other socioeconomic and clinical factors such as occupation and the presence of comorbidities, potentially skewing the outcomes by not capturing all influences on treatment adherence and retention. The reliance on existing records might have overlooked undocumented treatment interruptions or transfers. Secondly, some clients considered to have interrupted treatment may have reestablished care in other facilities without the service providers' knowledge, which may have affected the accuracy of the retention rates. Our analysis was limited by the small population of YLHIV who transitioned to adult care in tertiary health facilities and who initiated ART before the age of 20 years, which may narrow the generalisability of the findings. Also, the ongoing data cleaning in the country limited our analysis to 24 months. The small number of studies that reported 24-month treatment outcomes and studies in non-specialised or tertiary health facilities limited our ability to compare with other settings. The study did not adjust for some confounders, such as client experience and DSD models, which could impact the outcomes measured. Despite these limitations, our study had several strengths. This study had a much larger population of transitioned YLHIV than similar studies. This study is also one of the few studies that have reported on the treatment outcomes of adolescents transitioning from primary and secondary healthcare facilities and, to the best of our knowledge, is one of the first studies to report 24-month treatment outcomes post-transitioning to adult care in Nigeria.

### Conclusion

Continuity on treatment in adulthood is a recognised challenge for YLHIV. Our study found that a peer-led intervention to support YLHIV transitioning to adult care was associated with good continuity on treatment and undetectable viral load at 12 and 24 months post transition in relation to UNAIDS benchmarks.

This suggests that peer-based transition preparedness models can significantly enhance treatment outcomes for YLHIV in resource-limited settings. The evidence provided by this study will be useful to low and middle-income countries that provide ART services primarily through primary and secondary healthcare facilities. Policymakers should consider implementing these tailored, age-specific interventions to bridge the gap in service delivery during the critical transition phase.

This approach could help meet the UNAIDS targets for HIV control in resource-limited settings, where most YLHIV receive treatment [3,30,31]. Further investigation is needed to assess longer-term outcomes (up to 60 months) and to consider socioeconomic factors and the perspectives of YLHIV and healthcare workers on the transition preparedness model.

## Supporting information

**S1 Table: Dataset for the characteristics of YLHIV transitioned to adult care in 2018 and 2021 cohorts and their 12 and 24-months treatment outcomes.**
(XLSX)

## Acknowledgments

The authors acknowledge the healthcare workers and peer supporters for their selfless service and the Akwa Ibom and Cross River State governments for providing a conducive environment for project implementation.

## Author contributions

**Conceptualization:** Esther Nwanja.

**Data curation:** Uduak Akpan.

**Formal analysis:** Uduak Akpan.

**Methodology:** Esther Nwanja, Uduak Akpan, Chinazom Ekwueme, Ijeoma Uchenna Itanyi.

**Project administration:** Bala Gana.

**Supervision:** Ofonime Dixon-Umo, Echezona Ezeanolue, Augustine Idemudia, Onyeka Igboelina, Dolapo Ogundehin, Ezekiel James, Chika Obiora-Okafo, Olugbenga Asaolu, Bayo Onimode, Moses Katbi, Jemeh Pius, Omosalewa Oyelaran, Okezie Onyedinachi, Adeoye Adegboye, Andy Eyo.

**Visualization:** Uduak Akpan.

**Writing – original draft:** Esther Nwanja, Otoyo Toyo, Ogheneuzuazo Onwah.

**Writing – review & editing:** Esther Nwanja, Otoyo Toyo, Ogheneuzuazo Onwah, Godson Ndubueze, Onyeka Igboelina.

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
