## [Decision Letter · Decision Letter 0]

20 Nov 2024

PONE-D-24-33586Treatment outcomes among young persons living with HIV who transitioned to adult care in Southern Nigeria: A retrospective cohort studyPLOS ONE

Dear Dr. Nwanja,

Thank you for submitting your manuscript to PLOS ONE. After careful consideration, we feel that it has merit but does not fully meet PLOS ONE’s publication criteria as it currently stands. Therefore, we invite you to submit a revised version of the manuscript that addresses the points raised during the review process.

**ACADEMIC EDITOR:** The manuscript reports on the effectiveness of peer-led interventions in improving long term treatment outcomes for YLHIV who are transitioning to adult care in 2 states in Nigeria. The study is critical and timely due to challenges associated with adherence, retention, VL suppression among the sub-population. Therefore, it is important to be very explicit and to eliminate confusions in order to aid reproducibility and understanding. Hence, the authors should please respond to the following comments in addition to the comments from the 2 reviewers below:

**Financial disclosure:**

As a secondary data analysis, the data is likely to have been collected from a program funded by an agency (refer to methods – study setting – line 133). Please indicate this as part of your financial disclosure. You should include the agreement/contract number and indicate that the findings are exclusively the opinions of the authors and do not represent the position of the funder.

**Data availability:**

Not all data is presented within the manuscript. Therefore, indicate that data is available within the manuscript and other supporting documents, also provide rough data to support this as additional supporting document.

**Ethics statement:**

You indicated N/A. However, from line 233 you reported “ethical approvals/exemptions”. Please clearly indicate your ethics statement and not N/A.

**Some technical observations:**

Methods:

Please clearly and outline inclusion and exclusion criteria. though some of them have been mentioned under data collection, but there is need to explicitly outline them to avoid confusions. For example, the duration of the intervention; is it a client must be 1 year before data collection, or 8 months e.t.c.Line 203 – “transition was said to have happened after attaining 25 years”. Earlier it was stated that “transition assessment is conducted”. So, which of the two is more important? Attaining 25 years or passing transition assessment?Is there transition assessment checklist? Please provide key questions in the checklist or provide it's picture as a figure.Line 210 “retention in care disaggregated by….”. Retention in care is when someone satisfy the definition of “retention in care”. Consider using “continuity in care is disaggregated into….”.

Results:

Subsection “retention” – consider “continuity in care” instead. See earlier comment provided for line 210. Using “retention” the understanding is “client is retained”, and providing other categorization will cause confusion. However, retention can be a category under continuity in care, similarly retention with interruption, not retained, e.t.c can be other categories of continuity in care but not categories of retention.

Acknowledgement:

Consider acknowledging the funder.

We look forward to receiving your revised manuscript.

Kind regards,

Ibrahim Jahun, MD, MSC, PhD

Academic Editor

PLOS ONE

Journal requirements:    When submitting your revision, we need you to address these additional requirements. 1. Please ensure that your manuscript meets PLOS ONE's style requirements, including those for file naming. The PLOS ONE style templates can be found at https://journals.plos.org/plosone/s/file?id=wjVg/PLOSOne_formatting_sample_main_body.pdf and https://journals.plos.org/plosone/s/file?id=ba62/PLOSOne_formatting_sample_title_authors_affiliations.pdf 2. Please amend your authorship list in your manuscript file to include author Bala Gana.

**Reviewers' comments:**

Reviewer's Responses to Questions

**Comments to the Author**

1. Is the manuscript technically sound, and do the data support the conclusions?

Reviewer #1: Yes

Reviewer #2: Yes

2. Has the statistical analysis been performed appropriately and rigorously? 

Reviewer #1: Yes

Reviewer #2: Yes

3. Have the authors made all data underlying the findings in their manuscript fully available?

Reviewer #1: Yes

Reviewer #2: No

4. Is the manuscript presented in an intelligible fashion and written in standard English?

Reviewer #1: No

Reviewer #2: Yes

5. Review Comments to the Author

Reviewer #1: This manuscript highlights the effectiveness of peer-led interventions to improve long term treatment outcomes for youth living with HIV (YLHIV) who are transitioning to adult care. The YLHIV transitioning to adult care in the period following implementation of the peer-based transition preparedness model showed improved viral load suppression and retention to care, more significant at 12 months post transition, with a slight decrease at 24 months post-transition. Retention to care in this study, following implementation of the peer-based intervention, was concluded to be in line with the UNAIDS target for sustained ART, as well as other studies with similar peer-led interventions. Retention was further described under categories, which also highlights the challenges in HIV care in adolescence and youth. Additionally, the ART regimen is also described and discussed as a factor that may have contributed to the viral load suppression; this is particularly important in the context of the introduction of dolutegravir-based regimens at around the same time as implementation of the peer-based intervention.

A strength in this manuscript lies in the study population of youth and adolescents living with HIV, who have unique challenges within the scope of HIV care. Additionally, the setting of mostly in primary and secondary care facilities is also a strength, as most other studies reviewed focus on transition to care in predominantly secondary or tertiary facilities. The study presents a clear timeframe around the implementation of the peer-based intervention. A shortfall lies in the data set itself, as limited socioeconomic, geographic, and clinical factors were used in describing the potential factors influencing retention to care/ transition to care. Attrition of patients and exclusion of some patients also resulted in a much smaller sample size prior to the peer-based intervention compared to after. The adherence support platform is also an interesting component which was not fully discussed. The population was mostly female, but the reasons for this skew were not described in the text.

The areas of improvement are highlighted below:

Data presentation:

1) Line 270: 12 months percentages do not correlate with the data presented in Table 4. This is present in the abstract as well.

2) Line 276: retention at 12 months for females in 2018 percentage does not correlate with Table 5 data.

3) Table 7: Total number for column 12 months post transition, 2018 transition year, number with undetectable VL noted as 197/323 – correct denominator is 328. Percentage calculated is correct for a denominator of 328.

4) Adherence support platform data, parameters and results not discussed in main body of manuscript. I suggest potentially renaming this table as the adherence support platform data does not necessarily fall under “demographic characteristics”, but this is in reference to the VL suppression through the post-transition period/ year of transition.

5) Line 136: suggest adding that the bracketed numbers are “n=” as this can be misinterpreted as a reference.

6) Line 261 references Table 1 instead of Table 3 – requires a correction.

7) Line 267 references Table 4 after a period – suggest that is is referenced more clearly.

Technical clarifications:

1) I recommend changing the term “undetectable viraemia” to “undetectable viral load” which has been used in the body of the manuscript and better describes what the authors are referencing.

2) The Description of the Intervention (line 156) would be better linked to Table 2, and I would recommend removing Table 1 and incorporating this as descriptive text into the sub-headings of Pre-Intervention Period and Description of the Intervention.

3) Table 2 has a bracketed number at the end of “Transfer to adult care/service provider”: (25). Suggest clarifying if this is the age at transfer of “25 years”. I recommend deleting if this is reference.

References:

1) Vancouver referencing style is used throughout the relevant section, except for reference 3, 7, and 34; reference 50 appears to be incomplete.

2) Line 133 references the “2023 Spectrum Estimates”, but this is not included in the main references list.

3) Spacing text from reference brackets would help in readability within the main body of the manuscript. This also includes adhering to one convention when referencing an author in text, such as for lines 372 – 373.

Typos, spelling, grammar and phrasing issues

1) I recommend maintaining formatting as per the PLOS guidelines – these have been followed, but not consistently, as seen with the indentation (lines 39 and 40, but not 56 – this should be the paragraph, not the heading), table formatting and referencing within the text (including location of table, table headings, and table footnotes), and heading formatting (font size for main vs. sub-heading).

2) This manuscript will benefit from thorough copyediting, as typographical errors are present throughout the text, including spelling (example: line 24), inconsistent capitalization of words (some examples: line 97 – in, Table 3 + line 388 for “dolutegravir, Table 1 – “trained”, etc.), maintaining standardized hyphenation throughout the manuscript body (12-month vs. 12 month), and switching of verb tense (present vs. past tense).

3) Careful adherence to writing conventions needs to be maintained, such as numbering conventions at the start of sentences (line 146), and acronyms being expanded on first use in the text (see: ART, DSD).

4) Rewording confusing phrases such as lines 143 – 145 and lines 203 – 204.

5) Line 239: recommend the use of “Informed” consent instead of “Ethical” consent.

Overall, this study has strengths in its design, population and setting, but will benefit from a revision.

Reviewer #2: Treatment outcomes among young persons living with HIV who transitioned to adult care in Southern Nigeria: A retrospective cohort study.

1. Summary of the Research

The study examined the impact of a peer-based transition preparedness model on the treatment outcomes of young persons living with HIV (YLHIV) who transitioned to adult care. It compared the 12- and 24-month treatment outcomes of YLHIV who transitioned before and after the introduction of the model across primary, secondary, and tertiary health facilities in Southern Nigeria. The authors claim that the introduction of the peer-based transition preparedness model significantly improved the treatment outcomes of YLHIV transitioning to adult care with higher retention rates at 12 and 24 months post-transition for those who transitioned after the model was introduced, higher rates of undetectable viraemia at 12 and 24 months post-transition for the 2021 cohort compared to the 2018 cohort, and a greater proportion of YLHIV in the 2021 cohort had sustained undetectable viraemia compared to the 2018 cohort.

The study strengths include that: the sample relatively large, which enhances reliability and generalizability of findings; it used real-world data from medical records across multiple health facilities provides a realistic picture of the treatment outcomes; and it was conducted in a resource-limited setting where the majority of the global HIV burden live, making the findings highly relevant for similar contexts.

The weaknesses of the study include the following: it uses a retrospective methodology which may introduce biases related to data collection and recording; the analysis does not appear to adjust for potential confounders; the findings may not be generalizable to other regions or countries with different healthcare systems and support structures; and there is inadequate details on the peer-based transition preparedness model to better understand what specific components contributed to the improved outcomes.

My recommendation is that the manuscript needs minor revisions. The authors should consider providing more details on the peer-based transition preparedness model, including its components and how it was implemented. They should also discuss potential confounders and how they were addressed in the analysis or acknowledge this as a limitation if not adjusted for. Furthermore, they could consider performing a sensitivity analysis to assess the robustness of the findings. Finally, they could expand on the discussion to include potential mechanisms by which the intervention improved outcomes and how these findings can be applied to other settings.

Overall, the manuscript makes a valuable contribution to the literature on HIV care transitions and provides evidence for the effectiveness of peer-based interventions in improving treatment outcomes for YLHIV.

2. Examples and Evidence

2.1. Major Issues

2.1.1. The Abstract

The abstract provides a clear and concise summary of the study, including the background, methods, results and conclusion. It could be improved by addressing the following:

• As the peer-based transition preparedness model is central to the study, the authors could consider adding a sentence in the introduction to provide more context about the model.

• In the conclusion, the authors could highlight the potential implications of the findings for policy and practice to strengthen the impact of the conclusion.

2.1.2. Introduction

The introduction is well-written and provides a comprehensive overview of the context and significance of the study. It could however be enhanced by addressing the following:

• Provide more details on the peer-based transition preparedness model introduced in 2019, especially how it differs from previous models and why it was expected to improve outcomes considering the importance of transitioning YLHIV to adult care.

• More recent studies or reviews could be added to support the need for the current research to better position the study within the existing body of knowledge.

• The study rationale is not mentioned explicitly. It would help to clearly state the gaps in the literature that this study aims to fill.

• The introduction does not explicitly state the objectives of the study. I would suggest that a clear statement of the study's objectives be explicitly added to provide a better roadmap for the reader.

2.1.3. Materials and Methods

The methods section is detailed and describes the study design, setting, population, and data collection procedures. The use of logistic regression analysis is appropriate for comparing retention in care and undetectable viraemia between the two groups. However, the section would benefit by the authors addressing the following:

• In addition to the authors stating the study design as a retrospective cohort study, they could consider adding that it is a before and after study. Furthermore, they could provide the rationale for choosing the study design to strengthen the justification.

• Include a brief explanation of why the particular states were chosen. The authors also do not explicitly describe the sampling method for facilities and participants, and it could improve clarity if the manuscript explicitly states the sampling method used for both facilities and participants.

• The data collection process is described, but the researchers could clarify how missing data were handled and whether any imputation methods were used.

• The statistical methods used are appropriate, but the authors could provide more details on the logistic regression models, including how potential confounders were handled.

2.1.4. Results

The results are presented clearly, with a focus on the key outcomes of retention in care and undetectable viraemia. The use of tables to summarize the characteristics of the study population and the outcomes is effective. The results section could benefit from the following revisions:

• More detailed descriptions of the statistical analyses performed, which could include specifying the statistical tests used to calculate the p-values, the logistic regression models used, together with explicitly stating the covariates included in the models.

• Generally, the results narrative uses p-values to indicate significance, but it would be helpful to also report the effect sizes and confidence intervals for the estimates, as they provide more information about the magnitude and precision of the effects.

• The retention rates at 12 and 24 months are reported using p-values. However, the odds ratios for retention at 24 months seem to suggest a lower retention rate for the 2021 cohort compared to the 2018 cohort, which contradicts the statement that a higher proportion of those who transitioned in 2021 were continuously retained. This discrepancy should be addressed.

2.1.5. Discussion

The discussion interprets the findings in the context of the existing literature and the study's limitations. It provides a thoughtful analysis of the potential reasons for the observed improvements in treatment outcomes. It could however be strengthened by addressing the following:

• The authors could provide more details on the similarities and differences in study design, population and settings between their study and the cited studies to complement the comparison they have provided between their findings with those of the other studies.

• The discussion would be strengthened by a more detailed exploration of the potential mechanisms through which the peer-based transition preparedness model improves outcomes.

• Line 341 uses the word “disintermediation”. Since this word is not a common term in healthcare literature, it may be helpful to provide a brief explanation or definition of its use in the manuscript to ensure clarity for all readers, especially those who may not be familiar with its use in this context.

• The discussion on the higher retention rates at 12 months post-transition for the 2021 cohort is clear, but the lower retention rates at 24 months post-transition for the same cohort is not explained. The authors need to explore potential reasons for this decline in more detail.

• The authors have contextualized their findings within the broader literature on HIV care transitions but have not included a specific discussion on the unique challenges faced by YLHIV in Nigeria. This would provide more insight into the study's results.

• Though the limitations of the study are acknowledged, the authors could also discuss the potential impact of these limitations on the study's findings, e.g. how the use of secondary data might have influenced the results.

2.1.6. Conclusion

The conclusion succinctly summarizes the main findings, and it appropriately emphasizes the positive impact of the peer-based transition preparedness model on treatment outcomes. It can however be strengthened by addressing a number of issues.

• More discussion can be provided on the implications of the findings for policy and practice, particularly in low- and middle-income countries.

• The authors stated that "peer-led interventions to support YLHIV transitioning to adult care results in good treatment outcomes," but it does not specify what these outcomes are. It would be more informative to mention the key outcomes, such as retention rates and viral suppression rates, directly in the conclusion.

• The statement "irrespective of the facility type" is quite broad. While the study may have included various facility types, it's important to note any differences in outcomes between these types, if they exist. The authors could conduct analyses to explain the differences e.g. stratifying the data by facility type, calculating outcomes by facility type, doing statistical analysis to compare the outcomes between the different facility types, adjusting for confounders, etc.

• The conclusion does not suggest any future research directions. It would be beneficial for the authors to highlight areas where further investigation is needed, especially given the limitations mentioned in the discussion.

2.2. Minor Issues

2.2.1. Materials and Methods

• Though the peer-based transition preparedness model is provided in table 2, it could be helpful if the authors included a brief summary in the text.

• The authors mention ethical approval was obtained. It would help if they could discuss the ethical considerations in the methods section.

2.2.2. Results

• The viral load suppression data is well-presented in Table 6. I would suggest that the authors consider adding a note to explain the categories of viral load changes over the 24 months post-transitioning.

2.2.8. Language and Grammar

The language and grammar of the manuscript are generally clear and understandable. However, there are several areas where the language can be improved for clarity, conciseness, and readability. For example:

• "Thesa" instead of "These" in line 23.

• The sentence starting in line 62, “Undetectable viraemia was higher among the 2021 cohort ….” could be rephrased.

• Some long sentences could be broken down for better readability. E.g. the sentence starting "YLHIV is a heterogeneous group that includes adolescents (10-19 years) and youth (15-24 years) with both ….”

• It would help to use simpler language and words, and avoid uncommon words like “disintermediation”.

A thorough proofread of the manuscript could help identify and correct grammatical errors and typos, and improve some of the language for better readability and flow.

6. PLOS authors have the option to publish the peer review history of their article (what does this mean? ). If published, this will include your full peer review and any attached files.

**Do you want your identity to be public for this peer review?** For information about this choice, including consent withdrawal, please see our Privacy Policy .

Reviewer #1: No

Reviewer #2: **Yes: ** Brain C Chirombo, MBChB, MPH, MCPHP (Zim), FCPHP (ECSA), FAPH

---

## [Author Response · Author response to Decision Letter 1]

25 Jan 2025

Academic Editor

Financial disclosure: As a secondary data analysis, the data is likely to have been collected from a program funded by an agency (refer to methods – study setting – line 133). Please indicate this as part of your financial disclosure. You should include the agreement/contract number and indicate that the findings are exclusively the opinions of the authors and do not represent the position of the funder.

• Authors’ Response: We appreciate the editor's recommendation. However, this research did not receive specific funding, but we utilized data from a donor-funded project. We are unsure if this qualifies as a yes response to the financial disclosure question or if it's more appropriate to indicate that the authors received no specific funding for this work. We have revised the acknowledgement section to include the project funding source and agreement number as recommended. We have also indicated that the findings are exclusively the opinions of the authors and do not represent the position of the funder (lines 494-500).

Data availability: Not all data is presented within the manuscript. Therefore, indicate that data is available within the manuscript and other supporting documents, also provide rough data to support this as additional supporting document.

• Authors’ Response: We thank the editor for this recommendation. We have included rough data as additional supporting document and modified the data availability statement to reflect that data is available within the manuscript and other supporting documents.

Ethics statement: You indicated N/A. However, from line 233 you reported “ethical approvals/exemptions”. Please clearly indicate your ethics statement and not N/A.

• Authors’ Response: Thank you for highlighting this. The ethics statement has been added and now reads, "Permission to analyze the secondary program data was obtained from the Health Research Ethics Committee (HREC) in Akwa Ibom State on December 22, 2022 (HREC No. AKHREC/25/11/22/120), with an extension granted on January 9, 2024 (HREC No. AKHREC/5/5/23/149). Additional approval was obtained from the Office of International Research Ethics (OIRE) on February 24, 2023 (Project #: 2026278-1). Informed consent was waived because the study data was collected retrospectively, and only de-identified data were used for the analysis."

Methods:

• Please clearly outline inclusion and exclusion criteria. though some of them have been mentioned under data collection, but there is need to explicitly outline them to avoid confusions. For example, the duration of the intervention; is it a client must be 1 year before data collection, or 8 months e.t.c.

o Authors’ Response: We thank the Editor for this recommendation. The inclusion and exclusion criteria have been outlined as follows: (lines 209-217)

Inclusion criteria

YLHIV who turned 25 years old in 2018 and 2021.

YLHIV enrolled in the peer-based clubs in the 2021 cohort

Exclusion criteria:

YLHIV who transferred out of the facility after transition were excluded because their treatment outcomes could not be established during the follow-up period.

YLHIV with less than one year of HIV treatment before the transition were excluded from the 2021 cohort due to their limited exposure to the intervention.

YLHIV not enrolled in the peer-based clubs in the 2021 cohort.

• Line 203 – “transition was said to have happened after attaining 25 years”. Earlier it was stated that “transition assessment is conducted”. So, which of the two is more important? Attaining 25 years or passing transition assessment?

o Authors’ Response: We thank the Editor for this comment. Reaching 25 years of age was the most important criterion for transitioning. The transition assessment was informal and has been clarified in the manuscript. We have revised lines 192-194 to read “The second component was transition readiness assessment where healthcare workers routinely assessed YLHIV’s knowledge needs through informal discussions during clinic visits and club meetings and used this information to tailor support services.” Line 207-208 now reads “YLHIV were considered to have transitioned upon reaching the age of 25 years."

• Is there transition assessment checklist? Please provide key questions in the checklist or provide it's picture as a figure.

o Authors’ Response: We thank the Editor for this question. There was no formal transition assessment checklist. Assessments were conducted informally as part of routine club meetings. This has been clarified in the manuscript (lines 192-194). "The second component was transition readiness assessment where healthcare workers routinely assessed YLHIV’s knowledge needs through informal discussions during clinic visits and club meetings and used this information to tailor support services."

• Line 210 “retention in care disaggregated by….”. Retention in care is when someone satisfy the definition of “retention in care”. Consider using “continuity in care is disaggregated into….”.

o Authors’ Response: We appreciate this recommendation. “Retention in care” has been replaced with "continuity on treatment" throughout the manuscript.

Results:

• Subsection “retention” – consider “continuity in care” instead. See earlier comment provided for line 210. Using “retention” the understanding is “client is retained”, and providing other categorization will cause confusion. However, retention can be a category under continuity in care, similarly retention with interruption, not retained, e.t.c can be other categories of continuity in care but not categories of retention.

o Authors’ Response: We appreciate this recommendation. The authors have revised the manuscript, and "continuity on treatment" has been used to replace "retention in care" throughout the manuscript.

Acknowledgement:

• Consider acknowledging the funder.

o Authors’ Response: We appreciate the Editor for this recommendation. The authors have revised the acknowledgement section to include the funders. Lines 495-500 now reads, "We sincerely appreciate the peer supporters for their selfless service, the Akwa Ibom and Cross River State governments for providing a conducive environment for project implementation, and the generous funding and technical assistance provided by the American people through the US President’s Emergency Plan for AIDS Relief and the United States Agency for International Development."

REVIEWER 1 COMMENTS:

Data presentation:

1. Line 270: 12 months percentages do not correlate with the data presented in Table 4. This is present in the abstract as well.

• Author’s Response: We thank the reviewer for pointing out this discrepancy. The data has been corrected in all locations. Lines 60-63 of the abstract now read: "A higher proportion of those in the 2021 transition cohort was continuously retained both at 12 months and 24 months post-transitioning compared to those in the 2018 transition cohort (12 months: 96.7% vs 80.2%, p<0.001; 24 months: 92.7% vs 77.6% p<0.001)." The corrected data is also reflected in the manuscript on lines 290-293: "Also, a higher proportion of those in the 2021 transition cohort were continuously retained both at 12 months and 24 months post-transitioning compared to those in the 2018 transition cohort (12 months: 96.7% vs 80.2%, p<0.001; 24 months: 92.7% vs 77.6% p<0.001)."

2. Line 276: retention at 12 months for females in 2018 percentage does not correlate with Table 5 data.

• Author’s Response: We thank the reviewer for pointing out this discrepancy. The manuscript has been revised to ensure consistency between the in-text reporting and the table. Lines 297-299 now read: “As seen in Table 4, post-transition continuity on treatment at 12 months was higher among female YLHIV in the 2021 transition cohort compared to those in the 2018 transition cohort (97.4% vs 93.3%, OR: 2.71 95%CI [1.50 – 4.88]).”

3. Table 7: Total number for column 12 months post-transition, 2018 transition year, number with undetectable VL noted as 197/323 – correct denominator is 328. Percentage calculated is correct for a denominator of 328.

o Author’s Response: We thank the reviewer for pointing out this discrepancy. The authors have revised the manuscript and Table 6 now reflects 328 as the denominator for the 2018 cohort. (line 340-341).

4. Adherence support platform data, parameters and results not discussed in main body of manuscript. I suggest potentially renaming this table as the adherence support platform data does not necessarily fall under “demographic characteristics”, but this is in reference to the VL suppression through the post-transition period/ year of transition.

o Author’s Response: We appreciate the reviewer’s insightful comment regarding the adherence support platform data in Table 7. The data, parameters and results were not discussed in main body of manuscript because the authors did not intend to have the data presented. This is because we could not definitively establish client continuity on the adherence support platform over the 24-month period using the data available. This data has been removed from the table to ensure accuracy of the presented data.

5. Line 136: suggest adding that the bracketed numbers are “n=” as this can be misinterpreted as a reference.

o Author’s Response: We appreciate the reviewers for this comment. We have revised the manuscript to address this. Line 148-149 now reads: “The study was conducted in 155 PEPFAR-supported health facilities which include 104 primary, 49 secondary and 2 tertiary health facilities.”

6. Line 261 references Table 1 instead of Table 3 – requires a correction.

o Author’s Response: We thank the reviewer for identifying the incorrect table reference. We have corrected the reference to Table 2, as Table 1 was removed during this review process. The revised text on line 282 now reads: “This information is presented in Table 2.”

7. Line 267 references Table 4 after a period – suggest that is is referenced more clearly.

o Author’s Response: We appreciate the reviewer for this observation. We have revised the text to adhere to PLOS ONE submission guidelines, ensuring clear referencing of tables within the text. The revised text now reads (lines 285-287): “As shown in Table 3, post-transition continuity on treatment at 12 months was higher for those in the 2021 transition cohort…”

Technical clarifications:

1. I recommend changing the term “undetectable viraemia” to “undetectable viral load” which has been used in the body of the manuscript and better describes what the authors are referencing.

o Author’s Response: We appreciate the reviewer for this recommendation. We have implemented this change throughout the manuscript for consistency and clarity (lines 54, 64, 226, 231, 238, 320, 324, 332, 348).

2. The Description of the Intervention (line 156) would be better linked to Table 2, and I would recommend removing Table 1 and incorporating this as descriptive text into the sub-headings of Pre-Intervention Period and Description of the Intervention.

o Author’s Response: We appreciate the reviewer for this recommendation. The authors accept this recommendation. Table 1 has been removed from the revised manuscript and the relevant components of Table 1 are incorporated in the Pre-intervention Period and Description of the Intervention sections.

3. Table 2 has a bracketed number at the end of “Transfer to adult care/service provider”: (25). Suggest clarifying if this is the age at transfer of “25 years”. I recommend deleting if this is reference.

o Author’s Response: We appreciate the reviewer for this recommendation. We have included “years” as suggested to enhance clarity for the readers. The table now reads (Table 1), “Transfer to adult care/service provider (at 25 years).”

References:

1. Vancouver referencing style is used throughout the relevant section, except for reference 3, 7, and 34; reference 50 appears to be incomplete.

o Author’s Response: We thank the reviewer for this comment. We have revised the reference section to ensure that all references in the manuscript conform with the Vancouver referencing style.

2. Line 133 references the “2023 Spectrum Estimates”, but this is not included in the main references list.

o Author’s Response: We thank the reviewer for this comment. We have cited te source for the “2023 Spectrum Estimate” in the main reference list. It is now reference no. 23.

"Onovo AA, Adeyemi A, Onime D, Kalnoky M, Kagniniwa B, Dessie M, et al. Estimation of HIV prevalence and burden in Nigeria: a Bayesian predictive modelling study. EClinicalMedicine [Internet]. 2023 Aug 1 [cited 2025 Jan 4];62:102098. Available from: https://pmc.ncbi.nlm.nih.gov/articles/PMC10393599/"

3. Spacing text from reference brackets would help in readability within the main body of the manuscript. This also includes adhering to one convention when referencing an author in text, such as for lines 372 – 373.

o Author’s Response: Thank you for your suggestion to improve readability. We have added spaces before the reference brackets throughout the manuscript. We have also standardized the in-text citation format for consistency. For example, the referenced text in lines 429 – 431 now reads: "For example, Meloni et al (14), Griffith et al (48) and Rungmaitree et al (49) reported 34.6%, 67% and 73.3% viral suppression at 12 months respectively (14,48,49)…”

Typos, spelling, grammar and phrasing issues

1. I recommend maintaining formatting as per the PLOS guidelines – these have been followed, but not consistently, as seen with the indentation (lines 39 and 40, but not 56 – this should be the paragraph, not the heading), table formatting and referencing within the text (including location of table, table headings, and table footnotes), and heading formatting (font size for main vs. sub-heading).

o Author’s Response: We thank the reviewer for bringing the formatting inconsistency to our attention. The manuscript has been thoroughly reviewed and revised to ensure full adherence to PLOS ONE guidelines. This includes consistent indentation, correct table formatting (location, headings, and footnotes), and appropriate font sizes for main and sub-headings.

2. This manuscript will benefit from thorough copyediting, as typographical errors are present throughout the text, including spelling (example: line 24), inconsistent capitalization of words (some examples: line 97 – in, Table 3 + line 388 for “dolutegravir, Table 1 – “trained”, etc.), maintaining standardized hyphenation throughout the manuscript body (12-month vs. 12 month), and switching of verb tense (present vs. past tense).

o Author’s Response: We appreciate the reviewer’s comment and we have carefully reviewed the entire text and corrected the typographical errors, ensured consistent capitalisation, standardised hyphenation and maintained uniform verb tense in the manuscript.

3. Careful adherence to writing conventions needs to be maintained, such as numbering conventions at the start of sentences (line 146), and acronyms being expanded on first use in the text (see: ART, DSD).

o Author’s Response: We thank the reviewer for this recommendation. We have thoroughly revised the manuscript to adhere to writing conventions, ensuring that numbers are not used at the beginning of sentences and that all acronyms, such as ART and DSD, are expanded upon first use.

4. Rewording confusing phrases such as lines 152 – 155 and lines 207 – 208.

o Author’s Response: We thank the reviewer for this recommendation. We have carefully revised the manuscript to enhance readability by rephrasing confusing phrases. For example, lines 152–154 now read: “Tertiary health facilities, which serve as the highest level of referrals, have highly experienced experts offering specialized care for disease conditions” and lines 203–204 now read: “YLHIV were considered to have transitioned upon reaching 25 years of age”.

5. Line 239: recommend the use of “Informed” consent instead of “Ethical” consent.

o Author’s Response: We thank the reviewer for this recommendation, we have revised the ethics statement to replace "ethical consent" with "informed consent." (line 259) now reads: "Informed consent was waived because the study d

---

## [Decision Letter · Decision Letter 1]

20 Feb 2025

PONE-D-24-33586R1Treatment outcomes among young persons living with HIV who transitioned to adult care in Southern Nigeria: A retrospective cohort studyPLOS ONE

Dear Dr. Nwanja,

Thank you for submitting your manuscript to PLOS ONE. After careful consideration, we feel that it has merit but does not fully meet PLOS ONE’s publication criteria as it currently stands. Therefore, we invite you to submit a revised version of the manuscript that addresses the points raised during the review process.

**ACADEMIC EDITOR:** **Thank you for submitting your revised manuscript to PLOS ONE. You have adequately responded to comments from the Editor and the 2 Reviewers. However, your funding statement is still not acceptable. If fund have been allocated for the program that resulted in the data used for this study, then the study is considered funded. The funder does not necessarily need to participate or directly fund this study (manuscript). Therefore, we invite you to submit revised funding statement indicating the funder, the contract/agreement number. Also adjust your acknowledgement by removing any information about the funder.**

We look forward to receiving your revised manuscript.

Kind regards,

Ibrahim Jahun, MD, MSC, PhD

Academic Editor

PLOS ONE

**Journal Requirements:**

Reviewers' comments:

Reviewer's Responses to Questions

**Comments to the Author**

1. If the authors have adequately addressed your comments raised in a previous round of review and you feel that this manuscript is now acceptable for publication, you may indicate that here to bypass the “Comments to the Author” section, enter your conflict of interest statement in the “Confidential to Editor” section, and submit your "Accept" recommendation.

Reviewer #2: All comments have been addressed

2. Is the manuscript technically sound, and do the data support the conclusions?

Reviewer #2: (No Response)

3. Has the statistical analysis been performed appropriately and rigorously? 

Reviewer #2: (No Response)

4. Have the authors made all data underlying the findings in their manuscript fully available?

Reviewer #2: (No Response)

5. Is the manuscript presented in an intelligible fashion and written in standard English?

Reviewer #2: (No Response)

6. Review Comments to the Author

**Reviewer #2:**  (No Response)

7. PLOS authors have the option to publish the peer review history of their article (what does this mean? ). If published, this will include your full peer review and any attached files.

**Do you want your identity to be public for this peer review?** For information about this choice, including consent withdrawal, please see our Privacy Policy .

Reviewer #2: No

---

## [Author Response · Author response to Decision Letter 2]

12 Mar 2025

Academic Editor:

However, your funding statement is still not acceptable. If fund have been allocated for the program that resulted in the data used for this study, then the study is considered funded. The funder does not necessarily need to participate or directly fund this study (manuscript). Therefore, we invite you to submit revised funding statement indicating the funder, the contract/agreement number. Also adjust your acknowledgement by removing any information about the funder.

• Authors’ Response: We thank the editor for this recommendation. We have included the funder and the project agreement number in the cover letter as guided by the Straive Editorial Assistant of the journal. We have also removed all information about the funder in the acknowledgement.

---

## [Editor Report · Decision Letter 2]

14 Mar 2025

Treatment outcomes among young persons living with HIV who transitioned to adult care in Southern Nigeria: A retrospective cohort study

PONE-D-24-33586R2

Dear Dr. Nwanja,

We’re pleased to inform you that your manuscript has been judged scientifically suitable for publication and will be formally accepted for publication once it meets all outstanding technical requirements.

Kind regards,

Ibrahim Jahun, MD, MSC, PhD

Academic Editor

PLOS ONE
---

## [Editor Report · Acceptance letter]

PONE-D-24-33586R2

PLOS ONE

Dear Dr. Nwanja,

I'm pleased to inform you that your manuscript has been deemed suitable for publication in PLOS ONE. Congratulations! Your manuscript is now being handed over to our production team.

Kind regards,

on behalf of

Dr. Ibrahim Jahun

Academic Editor

PLOS ONE